

# Development of limb bone laminarity in the homing pigeon (*Columba livia*)

Rylee S. McGuire[1], Raffi Ourfalian[1,2], Kelly Ezell[3] and
Andrew H. Lee[1,3,4]

[1] Arizona College of Osteopathic Medicine, Midwestern University, Glendale, AZ, USA
[2] Kaiser Permanente Los Angeles Medical Center, Los Angeles, CA, USA
[3] Department of Anatomy, College of Graduate Studies, Midwestern University, Glendale, AZ, USA
[4] College of Veterinary Medicine, Midwestern University, Glendale, AZ, USA

## ABSTRACT

**Background:** Birds show adaptations in limb bone shape that are associated with resisting locomotor loads. Whether comparable adaptations occur in the microstructure of avian cortical bone is less clear. One proposed microstructural adaptation is laminar bone in which the proportion of circumferentially-oriented vascular canals (i.e., laminarity) is large. Previous work on adult birds shows elevated laminarity in specific limb elements of some taxa, presumably to resist torsion-induced shear strain during locomotion. However, more recent analyses using improved measurements in adult birds and bats reveal lower laminarity than expected in bones associated with torsional loading. Even so, there may still be support for the resistance hypothesis if laminarity increases with growth and locomotor maturation.

**Methods:** Here, we tested that hypothesis using a growth series of 17 homing pigeons (15–563 g). Torsional rigidity and laminarity of limb bones were measured from histological sections sampled from midshaft. Ontogenetic trends in laminarity were assessed using principal component analysis to reduce dimensionality followed by beta regression with a logit link function.

**Results:** We found that torsional rigidity of limb bones increases disproportionately with growth, consistent with rapid structural compensation associated with locomotor maturation. However, laminarity decreases with maturity, weakening the hypothesis that high laminarity is a flight adaptation at least in the pigeon. Instead, the histological results suggest that low laminarity, specifically the relative proportion of longitudinal canals aligned with peak principal strains, may better reflect the loading history of a bone.

## INTRODUCTION

Laminar bone is a form of fibrolamellar bony tissue in which the primary vascular canal network is organized into concentric interconnected layers (*Francillon-Vieillot et al., 1990*). It is dominated by circumferential vascular canals (*Currey, 1960*; *Francillon-Vieillot et al., 1990*; *De Ricqlès et al., 1991*; *De Margerie, 2002*; *De Boef & Larsson, 2007*;

---

Corresponding author
Andrew H. Lee, alee712@gmail.com

---

*Huttenlocker, Woodward & Hall, 2013*), which have elongated profiles in transverse view that run approximately parallel to the periosteal surface (Fig. 1). The proportion of laminar bone (laminarity (*De Margerie, 2002*)) in many adult avian species appears elevated in specific limb bones such the humerus, ulna, femur and tibiotarsus (*De Margerie et al., 2005*). Theoretical modeling suggests that these limb bones experience locomotor-induced torsion (i.e., by flapping in the humerus and ulna and by walking in the femur and tibiotarsus) (*Pennycuick, 1967*). Indeed, in vivo bone strain measurements confirm that torsional loading is substantial in the ulna of grounded but flapping turkeys (*Lanyon & Rubin, 1984*) and is dominant in the humerus of flying pigeons (*Biewener & Dial, 1995*). In addition, while walking, chickens and emus generate large torsional loads in the femur and tibiotarsus (*Biewener, Swartz & Bertram, 1986*; *Carrano & Biewener, 1999*; *Main & Biewener, 2007*). If these loading patterns are similar across birds, then the elevated laminarity observed in humeri, ulnae, femora, and tibiotarsi of many avian species may be a general feature of limb bones loaded habitually in torsion (*De Margerie et al., 2005*).

A purely biomechanical explanation for laminarity, however, remains problematic. Contrary to the laminarity hypothesis, the only two species with detailed measurements of flight-induced torsion (*Columba livia* (*Biewener & Dial, 1995*) and *Pteropus poliocephalus* (*Swartz, Bennett & Carrier, 1992*)) actually have negligible to low laminarity in the adult humerus (*Bennett & Forwood, 2010*; *Lee & Simons, 2015*; *Pratt et al., 2018*). Furthermore, any form of vascularization is substantially reduced in the superficial cortex, where maximum flight-induced torsional and bending loads are predicted (*Pennycuick, 1967*; *Carter & Spengler, 1978*; *Craig, 2000*). The relatively avascular parallel-fibered (or lamellar) bone of the superficial cortex reflects growth attenuation that occurs in birds and mammals as they approach adult-size (*De Margerie, Cubo & Castanet, 2002*; *Castanet et al., 2004*; *Ponton et al., 2004*; *Main, 2007*; *Kuehn et al., 2019*). Once adult size is reached, normal locomotor-induced loads are not able to stimulate deposition of new bone along the superficial cortex, laminar or otherwise (*Bennett & Forwood, 2010*). These observations suggest limits to where and when laminar bone can form. Therefore, further sampling across the development of the pigeon is needed to clarify the extent of laminar bone in juveniles.

Postnatal development in the pigeon is altricial (*Starck & Ricklefs, 1998*). Juveniles are flightless and nest-bound for most of the postnatal growth period (*Vriends & Erskine, 2005*; *Coles, 2007*; *Liang et al., 2018*). Only when nearly full-grown do they become powerful fliers (*Tobalske & Dial, 1996*). Thus, the pigeon is ideal to examine rapid structural compensation in the limb skeleton during locomotor transition. Because second moment of area at midshaft, which indicates bone wall thickness, increases disproportionately with body mass in the altricial-developing wings of the California gull (*Carrier & Leon, 1990*) and mallard (*Dial & Carrier, 2012*), we expect a similar increase in the altricial-developing forelimb and hindlimb bones of the pigeon. Specifically, polar section modulus, which is related to second moment of area and proportional to torsional strength and rigidity at midshaft (*Ruff, 2002*; *Young, Fernández & Fleagle, 2010*; *Ruff et al., 2013*), should scale with positive ontogenetic allometry. Furthermore, if laminarity is a reflection of locomotor-induced torsion (*De Margerie et al., 2005*), then it should increase dramatically with skeletal (and locomotor) maturity.

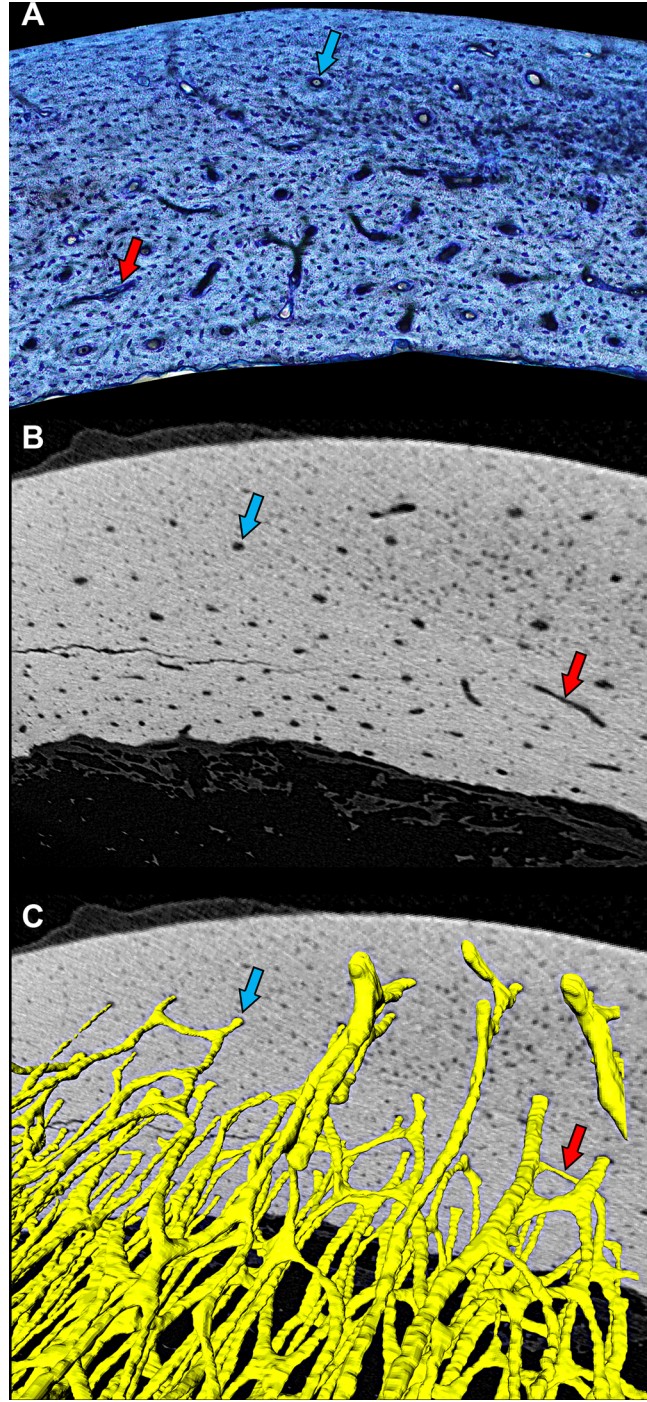

**Figure 1 Cross-sectional profile of a primary vascular canal is a useful approximation for canal orientation.** (A) The histological view of the dorsal octant of ulna from a post-fledge homing pigeon (MWU 257) was imaged from an undecalcified 100-μm transverse section stained with toluidine blue. The stain highlights edges of primary vascular canals at the plane of section, allowing accurate characterization of the cross-sectional profile of each canal. Canals with nearly circular profiles (aspect ratio < 3) have a longitudinal orientation (blue arrow), whereas those with greatly elongated profiles have a transverse orientation (red arrow). Whether the transverse orientation is specifically circumferential, radial, or oblique depends on how much the long axis of the canal profile is angled relative to the periosteal surface. This method of estimating three-dimensional orientation from two-dimensional canal

**Figure 1 (continued)**
profile assumes that canal shape is generally cylindrical. (B) MicroCT view (~2-μm voxel resolution) of the same specimen about 1 mm from the plane of histological section reveals circular (blue arrow) and elongated (red arrow) cross-sectional profiles of canals. (C) Three-dimensional rendering of canals (yellow) confirms that circular (blue arrow) and elongated (red arrow) profiles accurately reflect orientation—longitudinal and circumferential, respectively. MicroCT imaging was performed using a Zeiss Versa 520 at Arizona State University 4D Materials Science Center and was rendered with Avizo Lite (9.0.1).

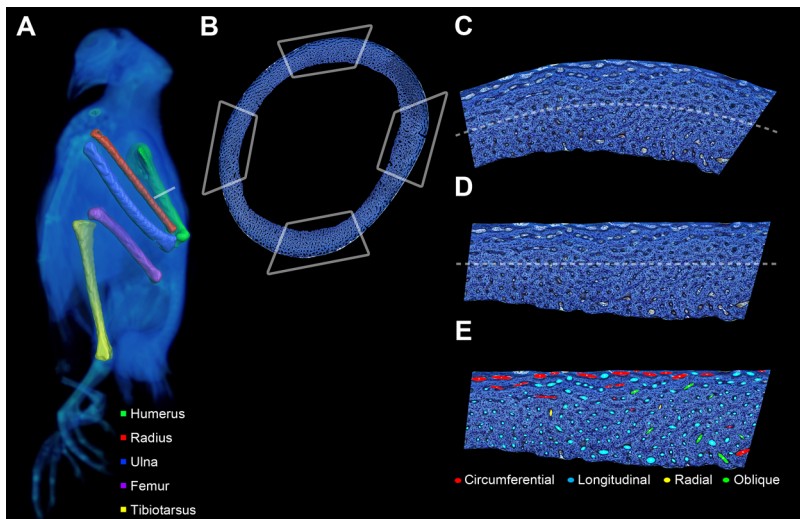

**Figure 2 Preparation of bone profile to evaluate laminarity index.** (A) The five listed bone elements were harvested from a growth series of 17 pigeons. Undecalcified and stained histological sections were taken from midshaft. For graphical illustration, MWU 269 was imaged using a Siemens SOMATOM Perspective CT scanner, and the left-sided elements were visualized with Avizo Lite. (B) Each histological section was divided into four octants representing cardinal anatomical positions based on a posture in which the pigeon is standing with the forelimbs extended laterally (i.e., cranial, caudal, dorsal and ventral for the wing elements; cranial, caudal medial and lateral for the hindlimb elements). Octant curvature (C) was straightened (D) using ImageJ. (E) Canals were fit with ellipses and classified based on orientation relative to the horizontal periosteal surface.

## MATERIALS AND METHODS

### Sampling and histology

We acquired salvaged carcasses of 17 homing pigeons (Stromberg's Chicks and Game Birds Unlimited; Pine River, MN, USA). The sample comprises a postnatal growth series of known mass (15–563 g) (Table S1). Although the age range of the sample is 0–9 weeks, the precise age of death for most individuals was not recorded. The right fore- and hindlimb of each individual was dissected to reveal the humerus, radius, ulna, femur, and tibiotarsus (Fig. 2A). The length of each element was measured (Tables S2–S6), and a 1-cm mid-diaphyseal block from each bone was excised using a rotary tool (Dremel 4000; Dremel, Mt. Prospect, IL, USA). We followed an established protocol for preparing plastic-embedded undecalcified bone (*Lee & Simons, 2015*). Specifically, two transverse 700-μm wafers were cut from each specimen at mid-diaphysis using a precision saw

(Isomet 1000; Buehler, Lake Bluff, IL, USA). Wafers were mounted (Gorilla Epoxy; Gorilla Glue Inc., Cincinnati, OH, USA) to glass slides and thinned to $100 \pm 10$ μm using a grinder/polisher (Metaserv 250; Buehler, Lake Bluff, IL, USA).

## Imaging

Sections were acid-etched and stained with toluidine blue (*Eurell & Sterchi, 1994*) to improve contrast of in-plane primary vascular canals (Fig. 2B). The stain also highlights secondary osteons (specifically cement lines) and their (Haversian) canals, which are traditionally excluded from measurements of laminarity (*De Margerie, 2002*). Whole section images pre- and post-staining were captured with a motorized microscope (Ni-U; Nikon, Tokyo, Japan) as previously described by *Lee & Simons (2015)*. Once imaging was completed, each section was mounted (Permount; Fisher Scientific, Hampton, NH, USA) with a glass coverslip (#1; Fisher Scientific, Hampton, NH, USA) for preservation (*Lee & Simons, 2015*). Sections are housed in the Arizona Research Collection for Integrative Vertebrate Education and Study (ARCIVES) at Midwestern University.

## Bone profiles

A bone profile was prepared from each montage following procedures described by *Lee & Simons (2015)*. Specifically, montages were sharpened in Photoshop (CS5; Adobe, San Jose, CA, USA) with the "Unsharp Mask" filter (5 px). The area bounded between the periosteal and endosteal surfaces was filled with white to represent bone. The surrounding non-bone area as well as in-plane vascular canals and resorption spaces were filled with black. Bone profiles were exported to ImageJ (1.51d; National Institutes of Health, Bethesda, MD, USA) for further analysis. We measured the periosteal circumference and vascular porosity (ratio of in-plane primary and secondary vascular area to total cortical area) of each bone profile (Tables S2–S6). Montages and bone profiles can be viewed at the Paleohistology Repository (*Lee & O'Connor, 2013*) and downloaded at Dryad.

## Ontogenetic scaling of polar section modulus

Bone profiles were imported into ImageJ, and the BoneJ plugin v1.4.1 (*Doube et al., 2010*) was used to calculate geometric properties such as second moment of area ($I$), polar moment of area ($J$), and polar section modulus ($Z_p$). These properties describe the distribution of material around the centroid of a given bone section and are inversely proportional to bending stress, overall (bending and torsional) stress, and maximum overall stress, respectively (*Schoenau et al., 2001*; *Ruff, 2002*; *Habib & Ruff, 2008*; *Young, Fernández & Fleagle, 2010*; *Hedrick et al., 2020*). In other words, a bone section with large $I$, $J$ and $Z_p$ experience less stress for a given load, giving it greater strength and rigidity to withstand larger loads before failure. Because those properties are closely related, we only present the $Z_p$ data (Tables S2–S6), which are the most relevant to test the torsional resistance hypothesis.

    $Z_p$ is proportional to the maximum torsional stress that occurs at the outermost surface of a bone loaded in pure torsion (*Craig, 2000*). It is calculated by taking the ratio of $J$ and the maximum distance between the centroid and the periosteal surface ($r_{max}$), where

**Table 1 Ontogenetic scaling of $\log_{10}$-transformed polar section modulus in the homing pigeon.** Parameters of linear regression fit are presented in $\log_{10}$ scale. Isometry is equivalent to a scaling exponent ($a$) of 1.

| Element | $R^2$ | $a$ | 95% CI | $b$ | 95% CI |
|---|---|---|---|---|---|
| Humerus ($n = 17$) | 0.93 | 1.76 | [1.58–2.46] | −3.52 | [−5.29 to −3.04] |
| Ulna ($n = 17$) | 0.90 | 1.80 | [1.46–2.59] | −3.93 | [−4.74 to −3.12] |
| Radius ($n = 17$) | 0.88 | 1.83 | [1.59–2.73] | −4.58 | [−6.86 to −3.92] |
| Femur ($n = 17$) | 0.87 | 1.59 | [1.34–2.61] | −3.43 | [−5.99 to −2.79] |
| Tibiotarsus ($n = 17$) | 0.95 | 1.39 | [1.28–1.82] | −3.02 | [−4.09 to −2.73] |

failure is most likely to occur. BoneJ calculates $Z_p$ without assuming circular or elliptical geometry by applying the following general relationship (Eq. (1)) to pixel data in the actual bone profiles:

$$Z_p = \frac{J}{r_{\max}} = \frac{\int r^2 dA}{r_{\max}},\qquad(1)$$

where $J$ is the integral sum of the area of each pixel $dA$ (2.36E−7 mm$^2$) representing bone that is a distance $r$ from the centroid, and $r_{\max}$ is the maximum distance between the centroid and the periosteal surface (*Doube et al., 2010*). $Z_p$ is an appropriate proxy for torsional strength and rigidity when the cross section of a long bone is nearly circular (*Craig, 2000*). To test the suitability of this proxy to each cross section, we used BoneJ to calculate the aspect ratio ($I_{\max}/I_{\min}$), which equals 1 in a circular cross section. Values for the bone sections range from 1.03 to 1.80 (Tables S2–S6). When compared to a reference figure (*Daegling, 2002*), the values of $I_{\max}/I_{\min}$ in our sample suggest errors in torsional rigidity less than 6.7%. Therefore, we find no major problem in using this proxy.

Using R (*R Core Team, 2019*), $Z_p$ and body mass were $\log_{10}$-transformed to linearize their relationship before performing separate Type I linear regression for each element. The scaling coefficients ($b$) and exponents ($a$) of the regression analysis are presented in Table 1. Because $Z_p$ and body mass ($M$) are each proportional to length$^3$, allometric scaling of the log-log model, $\log_{10}(Z_p) = \log_{10}(b) + a \log_{10}(M)$, was inferred if the 95% confidence interval of the scaling exponent ($a$) excluded the value of 1 (isometry). Note that we chose to use Type I instead of Type II (RMA) regression for two reasons. First, $Z_p$ and body mass were measured precisely with low error. Second and more importantly, there is natural individual variation in $Z_p$ for a given body mass. Such variation, whether influenced by intrinsic or extrinsic factors, is known to weaken Type II regression, which has a tendency to detect a steeper relationship than actually exists (*Kilmer & Rodríguez, 2017*).

### Laminarity index

We classified each vascular canal into one of four discrete categories of orientation (longitudinal, radial, oblique and circumferential). Although, the laminarity index (LI) was originally defined by *De Margerie (2002)* as the proportionate area of circumferential canals relative to the total area of vascular canals, we used a subsequent re-definition in which LI is the number of circumferential canals divided by the total number of canals

(*Simons & O'Connor, 2012*; *Legendre et al., 2014*; *Lee & Simons, 2015*). As such LI is a proportion and ranges from 0 (absence of circumferential canals) to 1 (ubiquity of circumferential canals).

We adopted a recently published method to quantify LI (*Lee & Simons, 2015*). Specifically, instead of counting every canal in an image of a bone section, we sampled LI from four anatomical octants and calculated mean LI representing approximately 50% of the total canals in a given section. The image of each section was divided into octants using Photoshop (Fig. 2B), and the four octants representing cardinal anatomical positions (i.e., cranial, caudal, dorsal and ventral for the wing elements; cranial, caudal, medial and lateral for the hindlimb elements) were extracted for analysis.

Canal orientation is measured relative to the local tangent to the periosteal surface. That surface, however, is curved in most bone cortices (Fig. 2C). Consequently, the local tangent varies across a curved cortex and requires repeated referencing to measure canal orientation. We used the method originally described by *Lee & Simons (2015)* to increase throughput and minimize error by straightening the curvature of each octant (Fig. 2D) using the "Straighten" function in ImageJ. Minimal distortion was verified by comparing octants overlaid with test angles before and after straightening.

We adopted the method by *De Boef & Larsson (2007)* to approximate the sectional profile of each primary vascular canal with a best-fitting ellipse using ImageJ. Aspect ratio and angle between the periosteal surface and major axis of ellipses were measured. To relate these measurements to canal orientation, we followed criteria originally proposed by *De Margerie (2002)*: (1) longitudinal canals have a roughly circular profile with an aspect ratio of less than 3; (2) circumferential canals have an aspect ratio greater than 3 with a major axis that is roughly parallel to the periosteal surface ($0° \pm 22.5°$); (3) radial canals have an aspect ratio greater than 3 with a major axis that is roughly perpendicular to the periosteal surface ($90° \pm 22.5°$); and (4) oblique canals have an aspect ratio greater than 3 with a major axis oriented between 22.5° and 67.5° to the periosteal surface (Fig. 2E). Each canal that branches was divided at the node, and the orientation of each subdivided canal was estimated using the methodology as described above.

The ellipse-fitting method is appropriate as long as canals are generally cylindrical. They tend to be in cortical bone (*Cooper et al., 2003*, *2011*; *Pratt & Cooper, 2017*), which ranges in vascular porosity from 0% to 30% (*Carter & Spengler, 1978*; *Zioupos, Cook & Hutchinson, 2008*). MicroCT inspection suggests this assumption is reasonable for avian cortical bone (Fig. 1). However, in cancellous bone (*Carter & Spengler, 1978*; *Zioupos, Cook & Hutchinson, 2008*) with vascular porosity greater than 30%, canals are too irregular to approximate with the ellipse-fitting method. Consequently, we measured canal orientation only in bone sections with porosity less than or equal to 30% (Tables S2–S6) and excluded the youngest specimens (MWU263, MWU 261, MWU 260 and MWU 267) from further analyses of laminarity.

## Robust principal component analysis and beta regression

In this study, we had to address the issue of multicollinearity among our explanatory variables: mass, bone length, and $Z_p$. Principal component analysis (PCA) enables the

**Table 2 Results from robust principal component analysis (PCA).**

| Element | | PC1 | PC2 | PC3 |
|---|---|---|---|---|
| Humerus | Eigenvalues | 5.384 | 0.100 | 0.022 |
| | Standard deviation | 2.320 | 0.316 | 0.148 |
| | Proportion of variance | 0.978 | 0.018 | 0.004 |
| | Mass eigenvector | 0.262 | −0.440 | 0.859 |
| | Length eigenvector | 0.522 | 0.813 | 0.258 |
| | $Z_p$ eigenvector | 0.812 | −0.381 | −0.443 |
| Ulna | Eigenvalues | 5.581 | 0.115 | 0.048 |
| | Standard deviation | 2.362 | 0.339 | 0.218 |
| | Proportion of variance | 0.972 | 0.020 | 0.008 |
| | Mass eigenvector | 0.274 | 0.896 | −0.350 |
| | Length eigenvector | 0.797 | −0.415 | −0.438 |
| | $Z_p$ eigenvector | 0.538 | 0.159 | 0.828 |
| Radius | Eigenvalues | 3.184 | 0.168 | 0.018 |
| | Standard deviation | 1.784 | 0.409 | 0.133 |
| | Proportion of variance | 0.945 | 0.050 | 0.005 |
| | Mass eigenvector | 0.453 | 0.689 | −0.566 |
| | Length eigenvector | 0.817 | −0.575 | −0.046 |
| | $Z_p$ eigenvector | 0.357 | 0.441 | 0.823 |
| Femur | Eigenvalues | 3.963 | 0.047 | 0.038 |
| | Standard deviation | 1.991 | 0.217 | 0.196 |
| | Proportion of variance | 0.979 | 0.012 | 0.009 |
| | Mass eigenvector | 0.348 | 0.845 | 0.407 |
| | Length eigenvector | 0.689 | −0.525 | 0.499 |
| | $Z_p$ eigenvector | 0.635 | 0.107 | −0.765 |
| Tibiotarsus | Eigenvalues | 6.706 | 0.233 | 0.048 |
| | Standard deviation | 2.290 | 0.483 | 0.218 |
| | Proportion of variance | 0.960 | 0.033 | 0.007 |
| | Mass eigenvector | 0.286 | 0.480 | −0.830 |
| | Length eigenvector | 0.867 | −0.499 | 0.010 |
| | $Z_p$ eigenvector | 0.409 | 0.722 | 0.558 |

formation of new, uncorrelated predictors (principal components) through linear combinations of the original variables. As such, we were able to resolve the issue of multicollinearity while still being able to assess the effect of each variable (*Fekedulegn et al., 2002*). PCA, however, is known to be highly sensitive to non-normal data. Therefore, we used robust PCA, which is appropriate for skewed data (*Hubert, Rousseeuw & Verdonck, 2009*), as implemented by the R package "rospca" (*Reynkens, 2018*). We standardized mass, bone length and $Z_p$ by median and median absolute deviation with the function "RobScale" (*Signorell, 2019*) in R. Data were grouped by homologous element, and a separate robust PCA was performed for each element (Table 2).

We used regression analysis to relate the minimum number of principal components (PCs) that account for at least 95% of the variation in the original variables with laminarity. However, LI values do not satisfy assumptions required of traditional linear regression because they are not normally distributed and are bounded between 0 and 1. To overcome these problems, we used the following beta regression model with a logit link function (Eq. (2)) to connect mean LI to the linear predictor:

$$\text{logit(LI)} = ln\left(\frac{\text{LI}}{1 - \text{LI}}\right) = \beta_0 + \beta_i \text{PC}_i + \ldots + \beta_k \text{PC}_k, \quad i = 1, \ldots, k \tag{2}$$

where logit(LI) is the logit link function for the mean of LI and the linear predictor is defined by $\text{PC}_i$, …, $\text{PC}_k$ as the scores of each principal component, $\beta_0$ as the intercept, $\beta_i$, …, $\beta_k$ as coefficients corresponding to each principal component, and $k$ as the number of principal components (*Ferrari & Cribari-Neto, 2004*). No 0-or 1-values were observed in laminarity, so we did not need to apply either a transformation to slightly shift boundary values or sophisticated zero-one inflated beta regression (*Smithson & Verkuilen, 2006*; *Douma & Weedon, 2019*). Instead, traditional beta regression was performed with the R package "gamlss" (*Rigby & Stasinopoulos, 2005*). To ease interpretation of fitted models on the scale of observed laminarity (0, 1), the linear predictor was transformed using the inverse logit function. Thus, the resulting expression (Eq. (3)) becomes a relationship between absolute changes in PC scores and mean LI:

$$\text{LI} = \frac{e^\eta}{1 + e^\eta}, \tag{3}$$

where the linear predictor $\eta$ is $\beta_0 + \beta_i \text{PC}_i + \ldots + \beta_k \text{PC}_k$ (*Douma & Weedon, 2019*). Standardized coefficients of the original predictors (mass, bone length and $Z_p$) were calculated by multiplying the vector of β-coefficients by the matrix of eigenvectors from the PCA (*Fekedulegn et al., 2002*: Eq. 23).

## RESULTS

### Histological description

At mid-diaphysis, the cortical bone in the limbs of the sampled pigeons become increasingly compact with growth (Tables S2–S6). In very young individuals ranging from 0 to 2 weeks (0–40% of adult mass), bone walls are largely cancellous (porosity > 30%) with irregular vascular spaces. That cancellous structure is consistent with rapidly-growing juvenile bone as seen in other avian species (*De Margerie et al., 2004*). Older individuals show compact bone with vascular canals. For each bone, peak laminarity (i.e., proportion of circumferentially oriented canals) occurs in pre-fledge juveniles (Figs. 2 and 3) that range approximately from 2 to 4 weeks of age and 40–70% of adult mass. These juveniles also show large disparity in laminarity with much greater values occurring in the humerus, ulna and femur than in the radius and tibiotarsus. In post-fledge individuals (4–9 weeks of age and 70–100% of adult mass), the outer bone cortex is poorly

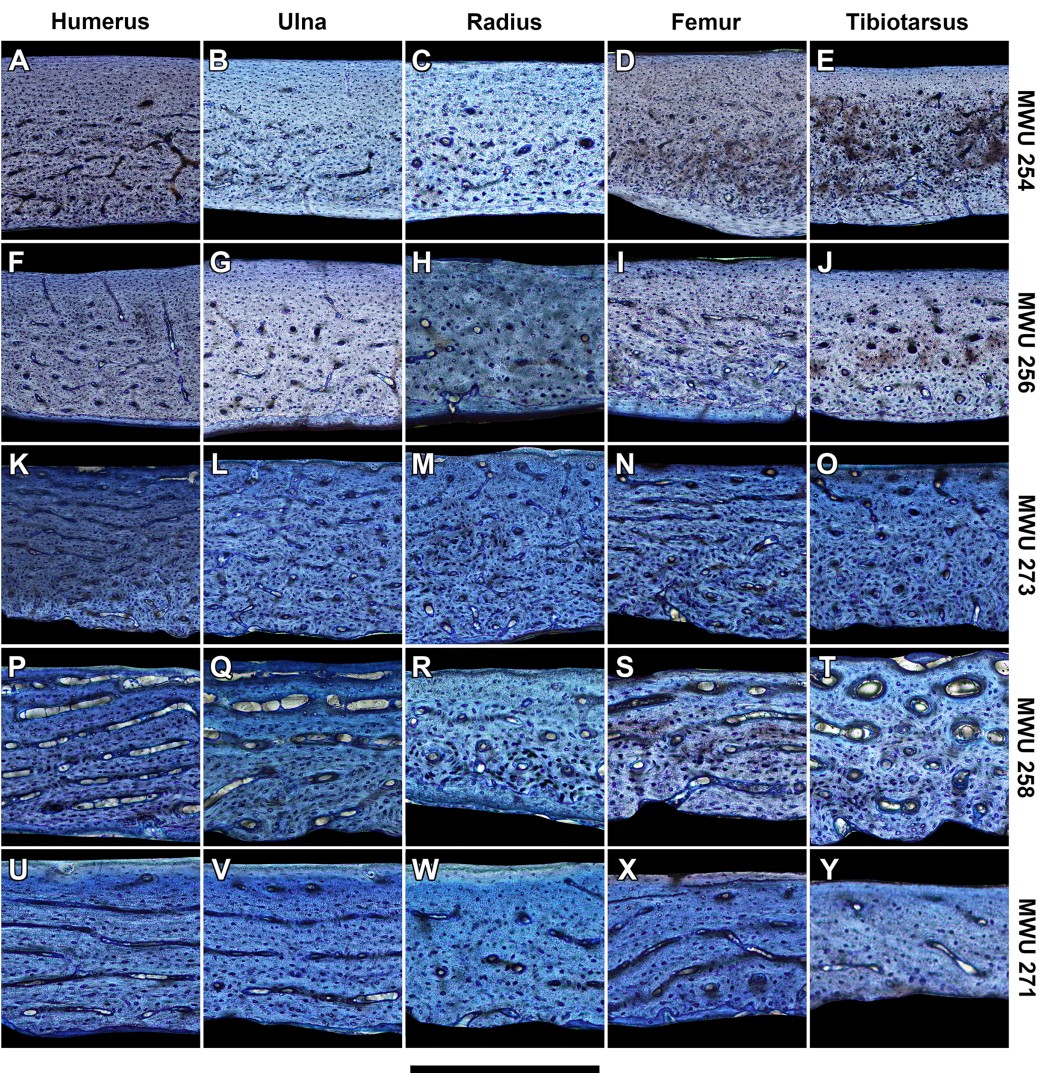

**Figure 3 Histology of representative bone elements from a growth series of homing pigeons arranged by mass.** In ascending order (bottom to top): MWU 271 (209 g, 3–4 weeks); MWU 258 (242 g, 2–3 weeks); MWU 273 (314 g, 4–5 weeks); MWU 256 (455 g, 5–6 weeks); MWU 254 (482 g, 8–10 weeks). Cortical bone porosity decreases with mass and age. Circumferential vascular canals are most abundant in the humerus, ulna and femur of pre-fledge juveniles (<5 weeks). Scale bar equals 600 μm (A, B, D–F and K), 480 μm (G, L and M), 400 μm (I, J, N–P), 343 μm (Q, T–V) and 300 μm (C, H, R, S and W–Y). Digital slides are freely accessible at http://paleohistology.appspot.com/Page/Columba_livia. html.

vascularized with a subperiosteal region of avascular parallel-fibered bone in the oldest individuals of the sample (Figs. 3A–3J). The remaining deep cortex is highly vascularized, but canals are predominantly longitudinal with only slight differences in laminarity among elements (Figs. 3A–3J). Secondary osteons are generally uncommon in the sample. The notable exception is the tibiotarsus in which secondary osteons are abundant in the deep cortex towards the end of the sampled growth period.

### $Z_p$ scaling analysis

The polar section modulus ($Z_p$) of the humerus, radius, ulna, femur, and tibiotarsus increases with growth (Tables S2–S6). When scaled to $\log_{10}$-transformed body mass, $\log_{10}$-transformed $Z_p$ shows significant positive allometry for all five sampled elements (Fig. 4) with 95% confidence intervals of the allometric exponents ($a$) each exceeding and excluding the isometric value of 1 (Table 1). The allometric exponent of the tibiotarsus is slightly (but not significantly) shallower than those of the other elements, in part reflecting the relatively large size of the tibiotarsus at hatch (Tables 1; Tables S2–S6).

### Robust principal component analysis

Robust principal component analysis is generally consistent across the five limb elements (Table 2). PC1 captures at least 95% of the variance in the original predictors: 98% for the humerus, 95% for the radius, 97% for the ulna, 98% for the femur and 96% for the tibiotarsus. We ignored the residual variance (approximately 2–5%) that is absorbed by PC2 and PC3, thereby reducing data dimensionality from three components to one. Mass, $Z_p$ (torsional rigidity), and bone length each have positive loadings on PC1. $Z_p$ and bone length have strong effects on PC1, but dominance varies depending on the element. In the humerus and femur, $Z_p$ is dominant or codominant with bone length, respectively, whereas in the remaining elements, length is dominant (Table 2). Taken together, the loadings are consistent with PC1 representing an ontogenetic axis. Small PC1 scores are associated with juvenile features (small mass with short bones that are relatively compliant to torsion), whereas large PC1 scores are associated with adult features (large mass with long bones that are relatively rigid to torsion).

### Beta regression

The fitted beta regression models with a logit link have significant β-coefficients (Table 3). They predict that an absolute one-unit increase in PC1 leads to a relative change by a factor of $\exp(\beta_1)$ in the ratio of laminarity (LI) to non-laminarity (1-LI). Thus, there is a relative decrease in the ratio of laminarity to non-laminarity by: 22% in the humerus ($\exp(-0.249) - 1$); 22% in the ulna ($\exp(-0.245) - 1$); 18% in the radius ($\exp(-0.197) - 1$); 22% in the femur ($\exp(-0.244) - 1$); and 13% in the tibiotarsus ($\exp(-0.137) - 1$) (Table 3). To better interpret these results as absolute changes in laminarity, the fitted beta regression model for each element was back-transformed with the inverse logit function and plotted. Each element shows a significant negative non-linear relationship between LI and PC1 (Fig. 5). Two groups are apparent. The first group, consisting of humerus, ulna, and femur, is characterized by relatively strong goodness-of-fit (pseudo-$R^2$ exceeds 0.70), relatively positive intercept (i.e., larger mean LI across ontogeny as calculated by Eq. (3)), and steep negative slope. In contrast, the second group, consisting of radius and tibiotarsus, is characterized by relatively weak goodness-of-fit (pseudo-$R^2 < 0.55$), relatively negative intercept (i.e., smaller mean LI across ontogeny), and shallow negative slope. Although laminarity generally decreases with ontogeny in the homing pigeon, laminarity in the radius and tibiotarsus may be influenced by additional unknown factors.

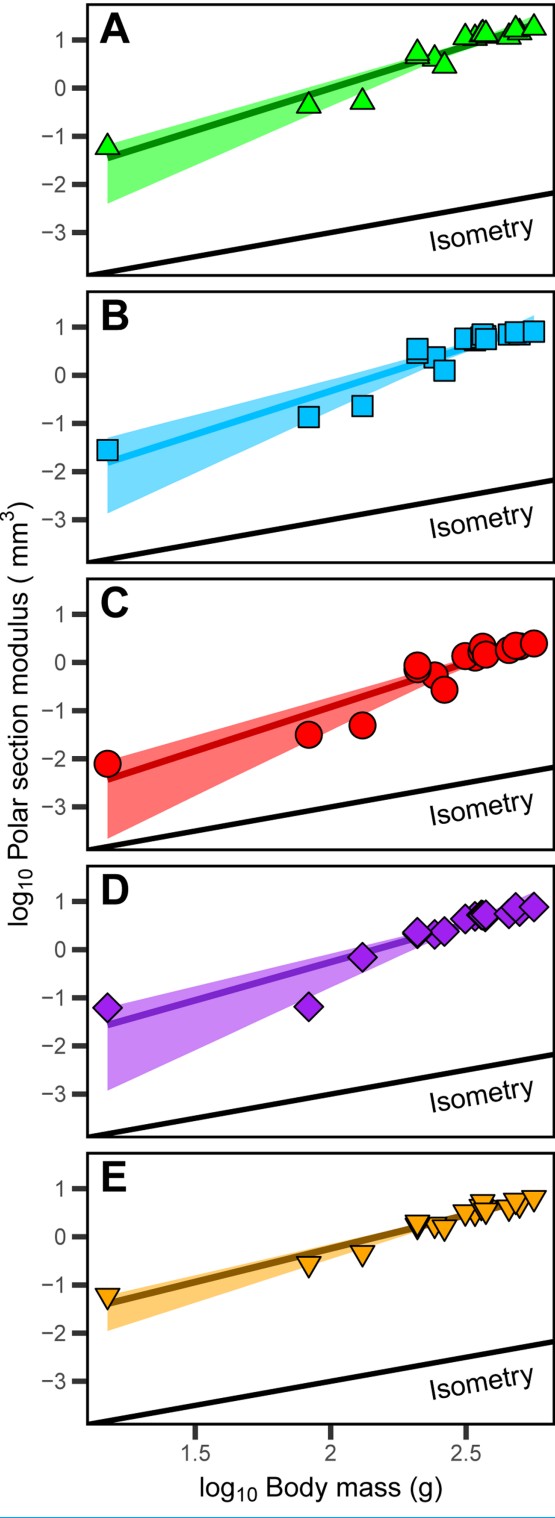

**Figure 4 Positive allometric scaling of polar section modulus ($Z_p$) in an ontogenetic series of (A) humerus, (B) ulna, (C) radius, (D) femur and (E) tibiotarsus from the homing pigeon.** Colored lines represent the relationship between $\log_{10}$-transformed body mass and $Z_p$ in the form: $\log_{10}(Z_p) = \log_{10}(b) + a \log_{10}(body\ mass)$. Shaded regions are 95% confidence bands. Solid black lines indicate the slope of isometry for reference.

**Table 3 Relationship between laminarity and scores of PC1 using beta regression with a logit link.** Standardized coefficients for each of the original variables (mass, $Z_p$, and bone length) are also listed.

| Element | Pseudo $R^2$ | $\beta_0$ | $p$-Value | $\beta_1$ | $p$-Value | Standardized Coefficients | | |
| --- | --- | --- | --- | --- | --- | --- | --- | --- |
| | | | | | | Mass | Length | $Z_p$ |
| Humerus | 0.726 | −1.283 | 9.5E−8 | −0.249 | 1.22E−4 | −0.065 | −0.130 | −0.202 |
| Radius | 0.440 | −2.585 | 5.7E−10 | −0.197 | 0.007 | −0.089 | −0.161 | −0.070 |
| Ulna | 0.852 | −1.670 | 2.0E−10 | −0.245 | 2.29E−6 | −0.067 | −0.195 | −0.131 |
| Femur | 0.819 | −1.564 | 4.9E−10 | −0.244 | 1.31E−5 | −0.085 | −0.168 | −0.155 |
| Tibiotarsus | 0.521 | −2.657 | 5.0E−11 | −0.137 | 0.002 | −0.039 | −0.119 | −0.056 |

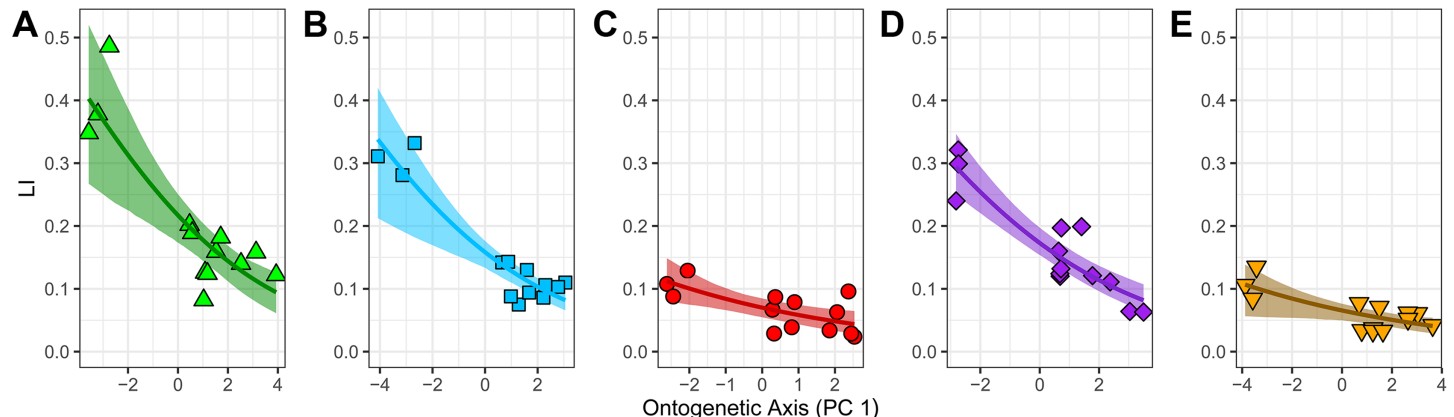

**Figure 5 Relationship between laminarity (LI) and the "ontogenetic axis" of variation.** Fitted beta regression models were back-transformed using the inverse logit function to ease interpretation of the ontogenetic trends in laminarity on the scale of 0–1. The back-transformed curves follow Eq. (3): $LI = \exp(\beta_0 + \beta_1 PC1)/(1 + \exp(\beta_0 + \beta_1 PC1))$. For each unit of increase along PC1 (the "ontogenetic axis"), laminarity decreases non-linearly given the following values of $\beta_0$ and $\beta_1$: (A) −1.283 and −0.249 in the humerus; (B) −1.670 and −0.245 in the ulna; (C) −2.585 and −0.197 in the radius; (D) −1.564 and −0.244 in the femur; and (E) −2.657 and −0.137 in the tibiotarsus.

To express the PC1 coefficient in terms of the original predictors (mass, length and $Z_p$), the eigenvectors and coefficient of PC1 were multiplied by each other (*Fekedulegn et al., 2002*: Eq. 23). The result is a set of principal component estimators of the standardized effects of the original predictor on laminarity (Table 3). In all but two elements, laminarity is most influenced by the effect of bone length, which is at least 1.5 times greater than the effects of either $Z_p$ or mass. Different patterns occur in the humerus and femur. In the former, the strongest effect is $Z_p$, and in the latter, the effects of bone length and $Z_p$ are similar in strength. Contrary to expectations, none of the standardized coefficients are positive indicating that, at least in the homing pigeon, growth in mass, bone length, and torsional rigidity have inverse effects on laminarity.

## DISCUSSION

### Positive allometric growth of torsional rigidity at midshaft

This study demonstrates that some structural changes in midshaft cortical bone are predictable responses to delayed locomotor development. Juvenile pigeons receive

extended parental care within protected nests and have limited mobility for over half of their postnatal growth period (*Levi, 1962*; *Janiga & Kocian, 1985*; *Johnston & Janiga, 1995*; *Vriends & Erskine, 2005*; *Liang et al., 2018*). Under those conditions, selection for relatively robust juvenile limbs is likely relaxed and reflected in "underbuilt" midshaft cortices. As juvenile pigeons approach adult size and become fully mobile, torsional rigidity increases rapidly in forelimb and hindlimb bones as indicated by significant positive ontogenetic allometry of midshaft $Z_p$ (Fig. 4).

Other volant birds also show positive ontogenetic scaling of geometric properties in forelimb bones. Like pigeons, larids and mallards hatch with altricial wings and are unable to fly until juveniles are nearly adult size. Late development of wings involves accelerated growth in pectoral muscle mass, wing surface area, and breaking strength of bones. The latter is attributed in part to disproportionately large increases to forelimb bone midshaft width and second moment of area (*Carrier & Leon, 1990*; *Bennett, 2008*; *Dial & Carrier, 2012*). Late investment in wings may allow resource allocation to develop other metabolically expensive organ systems first (e.g., digestive, nervous, cardiovascular, and integumentary) (*Carrier & Leon, 1990*; *Johnston & Janiga, 1995*; *Dial & Carrier, 2012*; *Altimiras et al., 2017*). A tentative conclusion from these results is that altriciality and positive allometric growth are coupled at least in avian wing development. However, we are aware that semi-intensively farmed Japanese quail from China show positive allometric growth of midshaft width in forelimb bones (*Ren, Wang & Zhang, 2016*), which is odd given the early locomotor capabilities of the wings. Positive allometry of midshaft width was also found in the femur and tibiotarsus, the latter being inconsistent with results from the closely-related chicken (*Biewener & Bertram, 1994*). Because the analysis on those Japanese quail used Type II regression, which has tendency to upward bias allometric slopes (*Kilmer & Rodríguez, 2017*), we suspect that negative allometry or isometry was mistaken for positive allometry. In any event, additional sampling is needed to test whether avian taxa with precocial wings capable of flight shortly after hatching (e.g., the Megapodidae (*Dial & Jackson, 2011*)) show negative allometric or isometric growth of forelimb bones, presumably to keep breaking loads relatively constant as they function in flight while still growing.

Unlike the pigeon, the California gull and mallard have hindlimbs that are functional for walking or swimming shortly after hatching. Their juveniles have relatively robust hindlimb bones and generally experience negative allometric or isometric growth in midshaft width and second moment of area, presumably to maintain locomotor performance and bone strength comparable to those of adults (*Carrier & Leon, 1990*; *Dial & Carrier, 2012*). The glaucous-winged gull also has functional hindlimbs at hatching. However, results from an ontogenetic study are not presented in terms of allometric scaling to body mass (*Hayward et al., 2009*). Instead, they show that sigmoidal growth in midshaft width in hindlimb bones peaks relatively early and is completed before forelimb bones. That pattern is consistent with negative allometry and supports the hypothesis that growth in limb bone geometry reflects locomotor needs during development.

Although distinct allometric growth trajectories of hindlimb bones may, to an extent, indicate locomotor ability in juveniles, there are at least two examples that caution

against oversimplification. Like previously discussed larids (*Carrier & Leon, 1990*; *Hayward et al., 2009*), the Black noddy shows negative allometric growth in second moment of area of the tibiotarsus (*Bennett, 2008*). However, the Black noddy is tree- or cliff-nesting, which is a behavior that provides refuge to juveniles but limits their locomotion during the growth period. Nevertheless, juveniles still grow tibiotarsii with relatively thick cortices that compensate for low mineralization to maintain bending strength and stiffness comparable to adults (*Bennett, 2008*). Allocation of resources to develop relatively robust locomotor-capable hindlimb bones in nest-bound juveniles is counterintuitive, and negative allometric growth in hindlimb bones may simply be a relic of ancestral selection for locomotor ability in juveniles shared among larids (*Bennett, 2008*).

Emus are capable of walking within two days after hatching (*Sales, 2007*). Yet femora and tibiotarsii of chicks and early juveniles tend to be "underbuilt" relative to adults as indicated by positive allometry in polar moment of area (*Main & Biewener, 2007*). This ontogenetic scaling relationship differs from the negative allometric growth reported in some precocial mammals (e.g., jackrabbit and goat), which increases bone safety factors in juveniles when performing locomotion similar to adults (*Carrier, 1983*; *Main & Biewener, 2004*, *2007*). The reason for the difference in hindlimb bone growth allometry among precocial species is not clear, but captivity is not likely a strong factor. Although emus and goats were studied under similar captive conditions with free access to exercise, they show divergent allometric patterns in hindlimb bone growth of polar moment of area (*Main & Biewener, 2004*, *2007*). Differences in behavioral ecology was noted as a potential explanation; when compared to young goats, juvenile emu spend more time foraging at slower speeds instead of evading threats at faster speeds. Therefore, the "underbuilt" cortices of hindlimb bones in juvenile emu may reflect relaxed selection for high-performance locomotion (*Main & Biewener, 2006*). In a similar way, juvenile pigeons may experience relaxed selection for torsionally rigid limb bones given that they delay development of aerial and terrestrial locomotion until nearly full-grown.

## Limb bone laminarity decreases with maturity

Contrary to previous work on domestic turkey (*Skedros & Hunt, 2004*) and emu (*Kuehn et al., 2019*), our results in the homing pigeon do not support the hypothesis that laminarity increases as a developmental response to locomotor-induced torsion. Unlike midshaft cortical geometry, which appears to respond to locomotor maturation by growing progressively rigid to torsion, laminarity decreases such that locomotor-capable adults have low values (2.4–15.8%; Figs. 3 and 5). Adults have low laminarity bone in part because the growth of the medullary cavity resorbs all record of earlier juvenile bone with higher laminarity. The remaining cortex with fewer circumferential canals (i.e., lower laminarity) reflects gradual reduction in periosteal deposition with skeletal maturation (Videos S1–S5). According to the laminarity hypothesis, low laminarity is a feature of limb bones adapted to locomotion involving reduced torsional loading. For example, low laminarity in the forelimb bones of birds with long and narrow wings may reflect habitual
loading in bending rather than torsion (*De Margerie et al., 2005*; *Simons & O'Connor, 2012*). The juvenile and young adult homing pigeons in our sample were raised in enclosed lofts with minimal crowding and free access to exercise, but without in vivo bone strain data, we recognize that biomechanical inferences based on our sample of homing pigeons remain speculation. Nevertheless, there is consilience between cross-sectional geometry from the current study and in vivo off-axis principal strains from other studies of adult birds, including feral pigeons (*Lanyon & Rubin, 1984*; *Biewener, Swartz & Bertram, 1986*; *Biewener & Dial, 1995*; *Carrano & Biewener, 1999*; *Main & Biewener, 2007*), that suggests substantial locomotor-induced torsion at least in the humerus, ulna, femur and tibiotarsus. A circumspect conclusion drawn from limited data is that bones adapted to resist torsional loading do not necessarily increase laminarity during development.

Recent work on adult feral pigeons reveals "high" laminarity at least in the humerus (*Skedros & Doutré, 2019*). The contrasting results suggest that laminarity in homing pigeons may not be as representative of the wild-type as assumed by the present study. However, the assessment of "high" laminarity in feral pigeons is based on casual inspection rather than a canal-by-canal count. Therefore, we cannot exclude the possibility that it is an overestimate. Indeed, when canal-by-canal counts are performed in other avian taxa either with histological (*Lee & Simons, 2015*) or computed tomographic methods (*Pratt & Cooper, 2017*), laminarity that is previously reported to be "high" changes to "low" (*Pratt et al., 2018*). Similarly, we expect a reassessment to find low laminarity in adult feral pigeons.

The focus on circumferential canals in the laminarity hypothesis needs better biomechanical justification. If vascular canals align circumferentially to reflect torsional loading on a given bone, then the canals should be roughly concentric with an approximate angle of 45° from the longitudinal axis. This angle corresponds to the orientation of peak principal strain during pure torsional loading (*Craig, 2000*). Empirical data from in vivo bone strain experiments on birds during locomotion demonstrate that the orientation of peak principal strain varies with limb bone and species but is still reasonably close to 45° (*Biewener, Swartz & Bertram, 1986*; *Biewener & Dial, 1995*; *Carrano & Biewener, 1999*; *Main & Biewener, 2007*). In contrast, circumferential canals are defined as "in-plane" features (*De Margerie, 2002*) that are aligned closer to the transverse plane of section. Because vascular canals are approximately cylindrical (Fig. 1), orientation can be estimated from a histological section by applying the in-plane aspect ratio of the canal (i.e., greater than 3: *De Margerie, 2002*) to a trigonometric equation (*De Boef & Larsson, 2007*)— angle in degrees = $180/\pi \cos^{-1}(\text{aspect ratio}^{-1})$. Using that equation, we calculate that circumferential canals are angled in excess of 45° and range between 70.5° and 90° from the longitudinal axis. This discrepancy suggests that circumferential canals are not as adapted to torsion as originally thought. Instead, canals once classified as "longitudinal" may be better torsion-resisting features. For example, we estimate that "longitudinal canals" with the aspect ratio of 1.41 would be aligned with peak principal strains in bone under torsion. The revised biomechanical interpretation may explain why "longitudinal canals" are so abundant not only in homing pigeons but generally in amniotes. Further testing is

needed but requires shifts from: (1) assumptions to experiments; (2) two-dimensional to three-dimensional imaging; and (3) categorical to continuous data.

The developmental approach used by the current study may inform how the nanostructural organization of collagen fibers also contribute to the torsional rigidity of bone. In adult birds, collagen fibers with oblique-to-transverse orientation (i.e., spiraling 45°–90° to the longitudinal axis) are especially abundant throughout the cortex of bones shaped to resist torsion (*De Margerie et al., 2005*). Similar collagen fiber orientation evolved independently in adult birds and at least one species of fruit bat (*Skedros & Doutré, 2019*) suggesting that it may be a fundamental adaptation of vertebrate flapping flight. If so, we expect collagen fiber obliquity and torsional rigidity of wing bones to increase with locomotor maturity. Preliminary evidence suggests that the predicted trend occurs at least in the ulna of growing turkey (*Skedros et al., 2003*; *Skedros & Hunt, 2004*). Future investigations should apply the developmental approach across a broader phylogenetic sample to test whether loading has similar effects on collagen fibers and vascular canals.

## CONCLUSIONS

The altricial limb bones of the homing pigeon show that some structural changes in midshaft cortical bone are predictable responses to delayed locomotor development. During postnatal growth, midshaft cortical geometry scales with positive allometry. Adults appear to have disproportionately stronger limb bones than juveniles consistent with the transition from limited to full mobility. Positive allometric growth is not exclusive to altriciality or precociality. Instead, it probably indicates reduced selection on locomotor performance in juveniles. As their limb bones develop midshaft cortical geometries that are increasingly rigid to locomotor-induced torsion, laminarity decreases. Because locomotor-induced torsion is likely greater in adults than juveniles, our results suggest that low laminarity may actually be a torsion-resisting feature. Bone with low laminarity contains abundant longitudinal canals that may be aligned with peak principal strains in bone loaded under torsion. Future directions include using microtomography and spherical statistics to assess how closely vascular canals approximate peak principal strain direction during growth.

## ACKNOWLEDGEMENTS

We thank Stromberg's Chicks and Gamebirds for access to their salvaged pigeons. Erin Simons provided access to shared lab supplies and equipment. Ravi Bhadriraju, who at the time was a research volunteer and federal work study student, prepared slides of the femur and tibiotarsus. Volumetric renders in Figs 1 and 2 were generated from microCT scanning performed by Baker Hughes Inspection Technologies and Arizona State University 4D Materials Science Center as well as from CT scanning performed by Sasha Willis at Midwestern University's Companion Animal Clinic. We thank Holly Woodward Ballard, Edina Prondvai, and John Skedros for their constructive reviews of the manuscript. This is Arizona Research Collection for Integrative Vertebrate Education and Study (ARCIVES) contribution no. 5.

### Funding

This work was supported by an Arizona College of Osteopathic Medicine Summer Research Fellowship to Rylee S. McGuire and intramural funds to Andrew H. Lee. The funders had no role in study design, data collection and analysis, decision to publish, or preparation of the manuscript.

### Grant Disclosures

The following grant information was disclosed by the authors:
Arizona College of Osteopathic Medicine Summer Research Fellowship.
Intramural Funds.

### Competing Interests

The authors declare that they have no competing interests.

### Author Contributions

- Rylee S. McGuire performed the experiments, analyzed the data, prepared figures and/or tables, authored or reviewed drafts of the paper, and approved the final draft.
- Raffi Ourfalian performed the experiments, analyzed the data, authored or reviewed drafts of the paper, and approved the final draft.
- Kelly Ezell conceived and designed the experiments, performed the experiments, authored or reviewed drafts of the paper, and approved the final draft.
- Andrew H. Lee conceived and designed the experiments, performed the experiments, analyzed the data, prepared figures and/or tables, authored or reviewed drafts of the paper, and approved the final draft.

### Data Availability

Two R scripts that contain the code necessary to replicate the analyses performed in this study are available as Supplemental Files.

Digital histological slides are available at the Paleohistology Repository (http://paleohistology.appspot.com/Page/Columba_livia.html) and digital copies of the slides are also available at Dryad: Lee, Andrew; McGuire, Rylee; Ourfalian, Raffi; Ezell, Kelly (2020), Development of limb bone laminarity in the homing pigeon (*Columba livia*), v5, Dryad, Dataset, DOI 10.5061/dryad.4b8gtht9c.

### Supplemental Information

Supplemental information for this article can be found online at http://dx.doi.org/10.7717/peerj.9878#supplemental-information.

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
