# Peer review of "Development of limb bone laminarity in the homing pigeon (Columba livia)"

_PeerJ, doi:10.7717/peerj.9878_

## Round 0.1 · original submission · Minor Revisions

Deer Andrew,
I have now received three reviews of your manuscript. Although the reviewers provide very positive feedback about your work, it is apparent that some aspects of the methodology are lacking in details and that the discussion could be deepened and broadened. Furthermore, I request you to double-check that you have taken due account of previous works (you may want to cite papers such as that of Skedros & Hunt, J. Anat., 205, 121-134, Pratt & Cooper, J. Anat, 233, 531-541, and others).

As per PeerJ policies (https://peerj.com/about/policies-and-procedures/#data-materials-sharing), all the raw data (including microCT scan data) have to be made available in a permanent public repository prior to formal acceptance.

Please, together with your unmarked revised manuscript, provide a marked-up copy as well as a document explaining how you have addressed each of the points raised by the reviewers.

Many thanks, and stay safe,
Fabien

·

Basic reporting

Well-written, figures are professional, graphs are easy to read. That the raw data, slide images, and R-code is made accessible is refreshing and appreciated.

Experimental design

Methods are sufficiently detailed and easy to follow.

Validity of the findings

Conclusions are well-stated and findings provide important information for the fields of histology and biomechanics. Most importantly, this research quantifies the results, including qualitative results from earlier studies.

Additional comments

This study provides important foundational work for histology and biomechanics, bringing up to date previous assessments and interpretations of bone laminarity. Rarely do I review manuscripts that are "ready to go" as-is, but this is one of them.

On line 55 there is a grammatical error, but that is all I have to comment on.

·

Basic reporting

Basic reporting criteria are all met.

Experimental design

There are some details I missed from the description of the Methodology which are essential for the reader to understand the results and their interpretations. I indicated all these in the General Comments part.

Validity of the findings

Findings seem to be sound but I'd need to see and understand all the missing details related to the data analysis to be sure.

Additional comments

This MS is a very valuable, succinct piece of work for our field, as it lays down important new stones for the foundation of our understanding on the locomotion-related ontogenetic osteohistology. Furthermore, I have to emphasize the scientific value (also for future studies) of the online source of section images this study provided which reveals an unexpected microanatomical and histological diversity not only through ontogeny and across different elements, but also among homologous elements of different specimens representing similar ontogenetic stages. As such, I strongly recommend this work for publication, albeit with some revision and consideration of a few potentially important issues I listed below.

My major concerns are related to some missing details of the methodology and the need for a more elaborated discussion on the dynamic changes of osteonal development that may to some extent alter the original vascular architecture throughout ontogeny. Finally, I suggest that the authors devote a few more and deeper thoughts to interpret ontogenetic histological patterns in the context of precocial vs altricial development by comparing their results with those of other cited papers on birds showing precocial hind limb development (see more detailed comment on this below for Line 271-273).

These comments are also highlighted in the annotated pdf I prepared for easier context identification.

Best wishes,
Edina Prondvai


Specific comments by lines

Introduction

Line 55: “…generate large torsional loads occur…”
Delete either “generate” or “occur”

Line 57: ”If these loading patterns are stereotypical across birds,…”
Are you sure that the word "stereotypical" is the right one to use here?

Line 64-68: “Furthermore…development”
But this is not really surprising as in adult flying pigeons the outer cortex is mostly avascular OCL, so it is not only short of circular canals but also of any vascular canals. Hence, it is obvious that vascularity will be restricted to the deeper cortex which represents only a late juvenile stage (as most of the juvenile cortex is resorbed to keep the low relative cortex thickness characterizing avian bones).

Line 75: “Skeletal compensation occurs”
What type of compensation? Please, specify.


Methods

Line 106-108: Please, state where these sections are housed.

Line 117-118: “and vascular porosity (areal ratio of in-plane vascular canals to total cortical area)”
Only primary or also secondary canals?

Line 127: “Imin/Imax,”
Please, indicate here what I stands for.

Line 132-133: “between log10-transformed Zp and the log10-transformed product of body mass and bone length.”
Were these parameters log10 transformed to make them normally distributed or to linearize their relationship? Please, specify.

Line 146-149: “The image of ….for analysis.”
Did you do additional analysis to compare different octants within one section and homologous octants of homologous elements among different specimens? Or you averaged out the LI among the four octants? It may still be interesting to check patterns among homologous octants as they pertain to anatomical directions and hence potentially different loading regimes.

Line 191-192: “Robust PCA was performed separately for each element (Table 2).”
How do you mean that for each element you did a separate PCA? Or you mean each homologous element groups separately, that is for all humeri together, for all ulnae together, etc? Based on the content of Table 1 and 2 that seems to be the case, but please, clarify.


Results

Line 208: “the limb bones of the pigeon become increasingly compact with growth”
I would say limb bone cortex/cortices to avoid any misunderstandings (e.g. increasing compactness could be interpreted as happening all over the mid-diaphysis, including the medullary cavity).

Line 214: ”aged 2–4 weeks (Figs. 2 and 3). As individuals mature (4–9 weeks of age),”
What does this age interval mean in terms of body mass % of adults? So which section of the pigeon growth trajectory does this phase represent? Related to this, it would be useful to show a general pigeon growth trajectory (age vs mass) with indications of the different ontogenetic stages represented in your sample, or these described hallmark ontogenetic stages where histology shows these differences.

Line 215: “bony tissue with poor vascularization along the superficial half to third of the cortical wall”
Would you call this part an outer circumferential layer? Is it composed of largely circumferentially oriented collagen bundles or in which direction do the fiber bundles run? Please, specify, as this can be interesting information for the discussion of torsion-related histology.

Line 220-221: “The polar section modulus (Zp) of the humerus, radius, ulna, femur, and tibiotarsus increases with growth”
I looked up the general calculation of this parameter and I found this:

Zp= (π(D4 – d4))/16D

If this is the one you used, your result would mean that the relative cortex thickness increases with growth (because if d [in this case the diameter of the medullary cavity] gets larger, Zp gets smaller and vice versa). Is this right? If yes, it's interesting, as I would have expected the other way around, that is that cortex gets relatively thinner with maturation (at least that's what I found in duck bones through ontogeny). Be this the case in pigeons or not, it would be useful to give the calculation method of Zp here or in the SI, maybe along with a small sketch of a tube and the parameters in the equation for easy comprehension, so that the reader can think for him/herself about what this and your results actually mean in terms of mid-shaft microanatomy / bone geometry through ontogeny.

Line 233-234: “In the humerus…. PC1”.
If I look at your Table 2, I find dominance of Zp in the humerus, equal effect of Zp and length in the femur and dominance of bone length in all the rest of the bones.

Line 249-250: “We converted the PC1 coefficient into standardized coefficients of the original predictors (mass, length, and Zp).”
What kind of conversion does this mean? You mean the coefficients of the linear combination of the variables giving PCs, so the variable loadings? Please, specify.


Discussion

Line 266-267: “As such, juveniles benefit from relatively robust skeletal proportions (negative allometry),”
Which parts of the skeleton? And relatively robust compared to what?

Line 271-273: “Moreover…2012).”
And what about leg bones? In this paragraph you only discuss wing bones, whereas it would be crucial to include how these findings in the wing bones relate to your findings in the hind limb bones. The entire skeleton following altricial growth trajectory, I'd expect to find similar allometries and growth-related structural changes in both, forelimbs and hind limbs. Which would then contrast the other birds that have precocial hind limbs but altricial forelimbs, like the mallard, seagull, etc, so the birds you referred to as well. Please, provide details on this, as this information is not only very interesting but very important in the interpretation of your data in this locomotor ontogeny context.

Line 291-292: “even when richly vascularized as in the pigeon, torsionally-loaded bone is not necessarily more laminar.”
Maybe not in later ontogenetic stages, but don't the humerus, ulna and femur show high laminarity early in ontogeny, and they consistently have higher LI than radius and tibiotarsus at any ontogenetic stage?

Line 313-315: “torsion-induced shear strain might have a minor effect on laminarity, but ontogenetic effects clearly dominate in bones selected for locomotion.”
Which ontogenetic effects? I mean, locomotion-related shear strain is also an ontogenetically changing effect in terms of its relation with the growing body mass and improving locomotor capacities. So what do you mean here exactly?

Line 318: “oblique-to-transverse orientation”
In relation to the bone's long axis? So oblique would be like spiraling at high angles to the long axis, and transverse would be circumferentially oriented fibers? Also, are you referring here to what we usually call in birds the OCL? So the outermost cortex layer? Or this applies throughout the entire adult cortex the inner half of which is still vascularized? Please, specify.


Conclusion

Line 329: “This developmental pattern differs from a recent report in growing emus,”
It does differ in some respects but corroborates in others, like the conclusion that laminarity is only weakly related to shear strain and that other ontogenetic factors play a more important role.

Line 331: “adaptation to locomotor-induced torsion involves elevated bone laminarity is not supported.”
To me, the picture is not as clear as that. You say in individual bones you find a decreasing laminarity from about 2.5 weeks posthatching up to adulthood. However, the pattern still seems to hold true that bones loaded more in torsion among limb bones (i.e. humerus, ulna and femur) show consistently higher laminarity throughout ontogeny within the same skeleton than the radius and tibiotarsus.
Furthermore, it is not only laminarity that decreases with ontogeny, but also vascularity as a whole, and the structural properties are then dictated mostly by large-scale collagen fiber orientation which is, if I understood it correctly, still dominated by circumferential bundles in the outer half of the almost avascular cortex. And this circumferential orientation might just as well reflect torsional load resistance in a cortex that gets more and more compacted during ontogeny.
Finally, I’m wondering whether you would consider the following scenario: During cortical osteonal compaction, which gives the cortex more structural rigidity in the locomotor maturation process, vascular canals that may have been circumferential originally start to get more and more filled up. In this process, the transversely/circumferentially running mid-sections of the canals which interconnect the longitudinal canals into the network referred to as laminar architecture are preferentially filled up and thereby leaving mostly longitudinal canals open. So this may dynamically change the overall vascular canal architecture into less laminar in the older cortical regions, i.e. in the inner cortex. Closing to adult sizes, the outer cortex becomes less and less vascular, being organized into an OCL.
If this was the case, it would explain why Skedros & Doutre "overestimated" laminarity in those bones based on their CPL images which, by showing collagen fiber orientation in osteonal infillings, still reflect the original but filled up circumferential vascular canal orientations.
In addition, looking at all of your images from the online source (which also made me realize the baffling diversity among a lot of homologous bones at similar ontogenetic stages), a pattern occurred to me that may support this scenario of osteonal development of laminar canals turning them into separated longitudinal canals: That is laminarity is usually more expressed in the outer cortical half (in juveniles where no OCL is yet present) or middle cortex (where OCL is oresent), while the inner cortical half shows a more longitudinal architecture. If laminarity was a permanent condition once it’s there, I’d expect to find more laminar bone in inner cortical halves at >2.5 weeks age, which I couldn’t really see in any of these elements when I checked all bones in your online image library. I’ve cut out an image of MWU 275 Ulna (4.5 weeks of age) to show you what I mean: here I zoomed in on an area where inner cortex shows more longitudinal canals, while some circumferential canals forming laminar cortical areas in the mid cortex are being “transformed” into separated longitudinal canals by osteonal infilling of the circumferential sections (sorry, this image doesn't appear here, but check your section).

Based on this pattern, I could hypothesize the following allocation system between growth and biomechanics:
1) Fast growing juveniles need high vascular supply in their limb bones but they are still affected by the torsional loads due to their "training" in the increasing locomotor maturation period. Thus, they will develop some laminar vasculature in the bones most affected by torsional "training" loads.
2) As the cortex grows radially, the innermost cortex gets more and more compacted, obscuring the once predominantly laminar vascular architecture, while the outer, i.e. newer cortex half keeps showing higher proportion of laminarity.
3) Closing to adulthood, growth rate decreases, less and less vascular canals are needed to sustain fast growth and biomechanial constraints become dominating. When the almost avascular OCL starts to be deposited and make up half of the adult cortex, collagen fiber orientation may reflect torsional load distribution within the skeleton (taking over from vascular canal indicators).

So all in all, I think it would be worth to consider these osteonal maturation processes that may change initial vascular architecture through time as well as the larger scale collagen fiber orientation in the OCL in your biomechanical interpretations.


Figure 1
Caption title: You mean that the shape of the vascular canals is cylindrical? Please, clarify.
Caption: Please, indicate the ontogenetic stage of the scanned specimen. Furthermore, what does figure panel 'c' represent? It's missing from this caption and it's not really clear based on the figure itself, either.
Also, figure panel labels indicated in the caption should correspond with figure panel labels in the figure itself. In this case either both capitalized or both lower case letters.
Finally, the two small insets in panels D and E which show virtual section planes look to me the same. It seems that it's only the view of the section plane that is different, i.e. you rotated the angle of the view. Please, explain it here more accurately.

Figure 2
Are anatomical directions in wing bones defined in extended wing position?

Figure 3
Could you indicate ages in weeks to the listed body masses of specimens as well?

Figure 4:
“Positive allometric scaling of polar section modulus (Zp)”
Allometric scaling of Zp with body mass * bone length? Why multiply body mass with bone length? They are already correlated independent variables. So what will that multiplication do? Won't it increase their multicollinearity to even higher degrees and thereby give a false relationship between Zp and these size measures? So how can this X axis be interpreted exactly? Please, explain. Also indicate here in the caption the underlying method of this figure (i.e. linear regression).

Figure 5
Which method did you use for fitting these curves on the LI vs PC1 scatter plot? And how do these graphs reflect the beta regression with the logit link function that you said you used for characterizing this LI vs PC1 relationship? So how are these figures related to the output of your beta regression? Please, give explanation in here as well as in the methods section so that the reader can interpret the link between the beta regression with logit link function and these figures, as well as the results of this analysis.

Table 1
Add that it's based on linear regression.
Also, here my question is the same as in the graphical representation of the same scaling relationship in Fig. 4: why did you multiply body mass with bone length for the independent variable? Wouldn't this inflate the effects of these correlating predictor variables?

Online image source of sections:
MWU 275: Can the tibiotarsus of this specimen record some sort of pathological condition? I’m just asking because of those large resorption cavities present mostly in the inner cortical half.

·

Basic reporting

General comments:

This is a robust, and very well written, study dealing with an important topic in bone developmental biology and biomechanics. The study is superbly crafted and conducted. I did not identify anything substantial with respect to problems or shortcomings in the methods and results. The authors have done an excellent job in describing their findings and have employed several different statistical approaches to evaluate and report the data.

I only have a few comments. My main concern deals with some of the conclusions, which can be readily modified.

Specific comments:

Lines 131-135: I would like to have you add 1-2 sentences regarding the rationale for why the ‘product of body mass and bone length’ was chosen. Provide a reference if available. Also state exactly why isometry is a slope of 0.75. Please make it clear to the less knowledgeable reader what the units are for both axes of Figure 4 so that the rise over run is 3 over 4 (correct?).

Lines 274-279: Skedros and Doutre could have done a better job in stating that higher laminarity is suggested in their images mainly in the outer one-half to two-thirds of many of their images. I was able to examine all of your sections that are available in the Paleohistology Repository. Additionally, I was also able to examine all images from the Skedros study, which are not accessible to the public, unfortunately. While their statement of “high laminarity” does appear to be overestimated, gross inspection of their unstained images generally resemble your MWU 273 image K. Therefore, I would not change what you have written, but I do wonder if your pigeons, especially the older ones, had access to flight activities similar to those used by Skedros and Doutre. In that study the pigeons were legally wild shot, and the first author shot several himself (this should have been reported in their Methods, but was not unfortunately). It would be presumed that pigeons that experience natural flight would have “sufficient torsion” in their humerus to warrant tissue-level adaptation (though, as argued below, this might NOT be expressed as variations in laminarity). Were your pigeons raised in cages, if so how large were the cases? I have made a telephone call to Stromberg’s Chicks and Game Birds in Pine River, Minnesota, but I decided to not wait for the answer given that this review was late anyway. Please make a comment on the flight activities (free ranging, caged, etc – size of cage or enclosure if known) of your pigeons, especially the older ones. Below, for other reasons, I have encouraged you to examine and include as a reference a study of histology of ulnae from young and older turkeys by Skedros and Hunt (2004 J. Anatomy, volume 205: 121-134). Note in that study the younger turkeys could actually fly for short distances even though they were a domesticated strain that were raised for consumption. These flying domesticated turkeys (these were the subadult/younger ones) were found to clearly have histology that correlated with a different load history (bending) of their ulnae than the ulnae of the older birds (presumably more torsion), which could not fly because of their body weight. How sure are you that your birds had sufficient NATURAL torsional loading to adequately test our main histological/laminarity vs. load history hypothesis?

Line 323: Examine Skedros and Hunt (2004, J. Anatomy) and note that they advanced the work of their 2003 paper. This 2004 study focuses more on regional variations and ontogenetic/ changes in collagen fiber orientation (CFO) data as being, perhaps, more sensitive than laminarity for identifying prevalent/predomination torsion/shear loading from bending loading. A similar conclusion regarding regional/load-history-related CFO variations is also stated in Keenan et al. 2017 (Am. J Physical Anthropology, vol. 162) – though this more recent study did not deal with birds or laminarity. The emphasis on CFO as a ‘preeminent material characteristic’ for interpreting load history in these two studies (2004 and 2017) leads me to the main criticism I have with your conclusion, as stated in your last sentence (see below in Validity of Findings):

Experimental design

This research is within the Aims and Scope of the journal.

The research question is well defined and relevant and meaningful.

This is a rigorous investigation and is performed at a high technical standard.

Methods are robust and seem reproducible.

Validity of the findings

Lines 334-335: You conclude that “This result is consistent with previous findings and suggests …. more strongly at the gross anatomical level rather than at the histological level.” This statement is premature because an adequate histological characterization of your bones has not yet been done/reported. Until analyses of predominant CFO are completed across your growth/age range, you should consider adding a new final sentence that is something like: “However, additional studies that quantify ultrastructural characteristics of our bones that are thought, when compared to laminarity, to more strongly correlate with and reliably distinguish the relative prevalence/predominance of torsion vs. bending (i.e., predominant collagen fiber orientation) are needed to confirm this conclusion.” This suggested sentence is perhaps too wordy, but please consider adding something about the fact that additional analysis of the bone MATERIAL is needed.

A final comment is in order, which I offer as food for thought. I was happy to see that you kept your focus on avian bone histology and did not try to generalize some of your findings/conclusions to other animals, especially to mammals that generally have appendicular bones with greater relative cortical areas in their mid diaphyses. But there is something more fundamental regarding your last 3 sentences, especially line 332 and lines 334-335, that I would like you to think about --- and perhaps you have anyway, so forgive me if I am being pedantic. In these final sentences you are conveying a very reasonable view that one should look closely at bone structure (“geometry”) when considering a bone’s load history. This seems very reasonable, and most readers would just think, in view of your data, ‘this seems correct and is very reasonable and consistent with conventional wisdom’. But I just would point out that you are dealing also with animals that have, generally, ‘low volume limb bones’. There is the idea that bones that might be constrained by being at the lower end of the range of the “stressed volume effect” – and typically do not remodel with secondary osteons perhaps, arguably, because of their relatively “low volume” -- would, ipso facto, seem to, or in fact do, adapt geometrically because material adaptability is not needed (this is discussed in more detail in a book chapter by Skedros 2012; Bone Histology, An Anthropological Perspective, ed. By Stout and Crowder). In contrast, ‘higher volume bones’ would adapt more readily at the material level because of the need to accommodate microdamage, though this MATERIAL-level adaptability could still be missed if the histological analysis was not comprehensive. Therefore, you are dealing with a paradigm of bone adaptability that seems to work well in avians of the size that you have studied, and you did not drift out of this paradigm. Nevertheless, it is plausible that your bones are adapting at the sub-microstructural level (I said “ultrastructural” above, others would say “nanostructural” level) in ways that are elusive because a ‘geometric/structural bias’ that permeates studies of low volume bones makes one think that there ‘is no more to look for’ in terms of what a bone might employ to effect tissue-level developmental adaptation. The likelihood that CFO and laminarity are dissociated in terms of downstream influences on bone tissue mechanics for local load history should be kept in mind, and I hope that you consider testing this hypothesis in the same bones of the present study. Having said this I was very happy to see that in your pigeon bone images shown in the Paleohistology Repository that you have listed CFO on a check list for future analysis.

Additional comments

Please see my general comments in the above sections of this review.

---

## Round 0.2 · Minor Revisions

Please, take into account the few comments of Reviewer 2. We can then move rapidly to Acceptance.

·

Basic reporting

All four criteria/areas are perfectly met. Additional figures and videos in the revised version increase the standards of the manuscript even above what I expected.

Experimental design

Revised MS meets all four listed criteria.

Validity of the findings

Revised MS meets all four listed criteria.

Additional comments

I am very pleased with this revision, actually beyond my expectations. I did have though a few minor questions and comments that I indicated in the annotated pdf of the revised MS. These are, however, only for the authors to consider before publication (maybe could be modified even at the accepted stage) and not major critical points.
I congratulate to the authors for their nice, high-standard, meticulous work.

Best wishes,
Edina Prondvai

---

## Round 0.3 · accepted · Accept

I am pleased to confirm that your paper has been accepted for publication.